# Association Between Social Networking Service Use and Body Image Among Elementary School Children in Japan

**DOI:** 10.3390/ejihpe15070125

**Published:** 2025-07-07

**Authors:** Asami Baba, Masumi Suzuki, Rikako Yoshitake, Yumiko Inose, Naomi Omi

**Affiliations:** 1Graduate School of Comprehensive Human Sciences, University of Tsukuba, Ibaraki 305-8574, Japan; s2330477@u.tsukuba.ac.jp (A.B.); s2230490@u.tsukuba.ac.jp (R.Y.); 2Gakuen no Mori Compulsory Education School, Ibaraki 305-0816, Japan; suzuki.masumi@mail.ibk.ed.jp; 3Department of Health and Nutrition, Faculty of Human Sciences, Tokiwa University, Ibaraki 310-0911, Japan; y-inose@tokiwa-u.jp; 4Institute of Health and Sports Sciences, University of Tsukuba, Ibaraki 305-8574, Japan

**Keywords:** body image, body size perception and preference, body size misperception, ideal body image, preadolescents, social networking services

## Abstract

The number of studies suggesting that social networking services (SNSs) use poses a risk to children’s body image continue to expand, but most studies have focused on adolescents. The study aimed to examine the associations between SNS use and body image among elementary school children in Japan. This study examined the relationship between SNSs use and body size perception and preference, body size misperception, and ideal body image among 1261 preadolescents (611 boys and 650 girls), aged 8–12 years (mean age = 9.64; SD =1.15; 52% girls), separately by sex. Using hierarchical multivariate linear regression analyses and logistic regression analyses, we examined body image factors and SNS use as the dependent and independent variables, respectively. Findings indicate that children who use SNSs do not significantly differ from nonusers regarding body dissatisfaction. However, SNS use is positively associated with body size misperception in girls. Additionally, for boys and girls, SNS use seems to increase the likelihood of admiring the body image of media figures rather than friends or classmates. Understanding how SNS use influences body image remains important for promoting healthy development in children.

## 1. Introduction

Media exposure has been linked to body image concerns among youth. Most studies have focused on exposure to mass media formats such as television programs, magazines, and advertisements. However, given the extensive SNS use by young adults ([12]), it has become increasingly important to study the potential relationship between SNS use and body image concerns among young boys and girls. SNS use has been associated with problematic behaviors related to actual body image, such as excessive food restriction and investment in appearance ([33]; [69]). In particular, social comparison use has been associated with higher levels of bulimic symptoms and eating disorders ([64]). Therefore, identifying associations with SNS use is critical in addressing issues related to body image.

### 1.1. SNS Use and Body Size Perception and Preference

Social networking services (SNSs) considerably influence body image. Increased SNS use has been linked to body image disturbances and eating disorders in youth and adults, with small to moderate effect sizes ([18]; [20]; [58]; [73]). Recent studies have also suggested that higher media dependence is associated with body image disturbances in adolescents ([26]). Moreover, SNSs addiction was found to be associated directly with eating disorders and indirectly with poor body image in adolescents and young adults ([40]). Additionally, heavy SNS use has been associated with body size perception and preference among Australian preadolescents (mean age = 12.76; [57]). Thus, SNS use is a factor that affects body size perception and preference and body image disturbance in young people. Forming a proper body image at an early age is crucial for long-term healthy body development. However, the effects of SNS use on body image disturbance extend beyond body size perception and preference.

### 1.2. SNS Use and Body Size Misperception

SNS use also seems to affect body size misperception. Body size misperception is the discrepancy between one’s perceived weight and one’s actual weight status and is the result of a perceptual disorder that does not accurately assess one’s body size, combined with body dissatisfaction, including negative feelings and cognitions about one’s body ([31]; [19]). A study of Korean middle and high school students found that SNS dependence contributed to increased body size misperception ([3]). While studies on SNS use and body size misperception remain limited, these distortions can have serious physical implications. Incorrect body size misperception can lead to greater body size perception and preference. Body size misperception may be associated with eating disorders, unhealthy weight control behaviors, and mental health issues, including depression ([19]). In turn, this may lead to the belief that one is overweight or underweight despite having a normal weight, excessive weight loss concerns, and an increased risk of eating disorders (e.g., overeating and underrating). Conversely, improving body size misperception may reduce body size perception and preference and social comparison, fostering healthier eating and self-care behaviors. Further investigation into the relationship between SNS use and body size misperception is necessary.

### 1.3. SNS Use and Ideal Body Image

Furthermore, the rapid expansion of SNS use has reshaped the ideal body image (IBI) standards. In the 2000s, before SNSs became a dominant information source, appearance-related pressures were believed to stem from the following three main sources: family, peers, and traditional media. These pressures would cause adolescent girls to internalize a thin appearance as ideal and make social appearance comparisons, resulting in body dissatisfaction ([71]; [60]). Family and friends’ dieting behavior could influence them ([11]). Traditional media, including television and fashion magazines, promoted the desire for thinness in young women through their portrayal of thin models and celebrities ([1]). However, in recent years, adolescent girls’ main information sources have shifted from television to SNSs ([10]; [37]). This suggests that the highly visual SNSs may influence the overestimation of body image, the fear of weight gain, and the obsession with body image, promoting eating disorders ([61]). Viewing appearance-focused content and negative online social comparisons on SNSs indicates increased body dissatisfaction ([8]). A study found that people are more likely to internalize a thin-ideal image when comparing their appearance with others of the same gender on SNSs than when comparing themselves to friends or with models on TV or in fashion magazines ([18]). This suggests that SNS-based body comparisons, rather than those involving close family or friends, may reinforce the desire to lose weight and reinforce restrictive eating. Consequently, for many youths, the IBI may be shaped by people on SNSs.

### 1.4. Differences Among Girls and Boys

The impact of SNS use on body image is particularly pronounced in girls. Compared to boys, girls use SNSs at higher rates ([6]) and face greater risks of body image disturbances ([44]) compared to boys. Generally, appearance is more important in girls’ peer relations than among boys ([70]), and girls are reported as more susceptible to interpersonal influences from parents and peers than boys ([62]). Exposure to idealized body images on SNSs has been associated with lower body satisfaction in girls, likely because of social comparison processes ([2]). Additionally, earlier puberty onset in girls than in boys ([46]) may contribute to significant sex differences in early body image assessment. Recently, concerns have been raised about assessing body image in adolescent boys to capture key aspects of boys’ body image ([16]). Many studies confirm increased body size perception and preference in girls, whilst its prevalence and patterns in boys remain less clear.

However, recent trends indicate a rising prevalence of thinness among preadolescent boys in Japan ([36]). In Japan, 1.3% of boys aged 8–9 and 2.6% of those aged 10–12 are categorized as thin, compared to 1.6% and 2.8% of girls in the same age groups, respectively. Since 1977, the proportion of underweight girls has remained stable, whereas rates have increased among boys ([36]). Globally, adolescent boys also experience body dissatisfaction and eating disorders symptoms ([52]) and frequently use SNSs ([6]). Additionally, preliminary reports suggest that body image issues are significant concerns for boys ([43]; [50]), and boys experience pressure to lose weight ([59]). Given that thinness and body size misperception are key risk factors for eating disorders ([38]), body image disturbances represent a significant public health concern in Japan. Furthermore, boys generally have more of the ideal of body muscularity compared to girls ([49]; [53]). Even with the same thinness judgments, there may be differences between boys and girls in their ideals for body image. Therefore, efforts should focus on fostering a healthy body image before adolescence and education to prevent unhealthy dieting for both sexes.

### 1.5. Preadolescent Media Use

Although previous research has primarily focused on adults, relatively little attention has been given to preadolescents. Whilst substantial evidence links SNS use to body image disturbances and eating disorders ([18]; [20]; [51]), studies have specifically examined relationships between SNS use and body image in preteen samples. However, although SNS use increases rapidly from age 12 onwards ([63]), the number of preadolescent SNS users has surged globally. In Japan, 98.1% of schoolchildren in grades 4–6 (ages 9–12) use digital media, and 58% have reported using SNSs such as LINE (48%), TikTok (31%), Instagram (9%), and X (2%) ([39]). The average daily screentime for this age group is 167.7 min, with 41.4% of them having 3 or more hours of screentime daily ([23]). These findings suggest that a substantial proportion of Japanese children engage with SNSs before adolescence. Given the annual surge of screentime use yearly and the decreasing age of SNS users, SNSs now constitute a critical environment for children today. Because much of children’s social and emotional development occurs online, SNSs exert a much greater influence. However, most children possess limited self-regulation skills and remain vulnerable to peer pressure, increasing their risk of the negative outcomes associated with SNSs ([40]).

Both boys and girls have reported body size perception and preference before preadolescence ([15]). These patterns indicate that SNS use may contribute to body image disturbances even before adolescence. Therefore, examining how SNS use relates to body image, including sex differences, is essential for addressing these issues in preadolescents.

### 1.6. The Current Study

The relationship between SNS use and body image has been examined in adolescence but has not been tested in preadolescence. The study aimed to examine the associations between SNS use and body image among preadolescent girls and boys. Specifically, this study explores its relationship with body size perception and preference, body size misperception, and IBI.

According to previous research examining the association between SNS use and body size perception and preference in adolescents, SNS use is a factor that affects body size perception and preference. This study addresses the understudied area of SNS use and body image among preadolescent children. By providing a more detailed examination of body image and exploring sex differences, it represents an initial effort to bridge the gap in previous research. Even in preadolescents, it may be hypothesized that children who use SNSs more frequently will be more dissatisfied with their bodies than children who use SNSs less frequently (Hypothesis 1). Previous research also indicates that incorrect body size perception may lead to greater body size perception and preference. It is thus hypothesized that children who use SNSs will have a greater size misperception towards body image (Hypothesis 2). Furthermore, because it could be predicted that IBI is also changing with the spread of SNSs, it could be hypothesized that children who use SNSs will have an IBI of media figures instead of their friends and classmates (Hypothesis 3).

## 2. Materials and Methods

### 2.1. Participants and Procedures

This cross-sectional study included 1261 third- to sixth-grade preadolescents (611 boys, 650 girls) aged 8–12 years (mean age = 9.64; SD = 1.15, 52% girls) from two urban public schools in Japan. Data were collected using a self-reported questionnaire. To obtain approval, school principals were first contacted and provided with information on the study’s aims and questionnaire content. Once they granted permission, class teachers then distributed documents to parents stating the study’s purpose, methods, data protection measures, results dissemination, and voluntary participation, asking for written consent for their schoolchild’s participation. Finally, participants gave digital consent by selecting either “I agree to participate” or “I do not agree to participate.” The privacy rights of participants have been observed and informed consent was obtained from the parents and preadolescents. If they chose not to participate, participants’ decisions were acknowledged and the questionnaire was then terminated. Additionally, participants could also withdraw at any time by closing the browser without their answers being recorded.

The questionnaire was administered via self-report Google Forms on tablets. Data collection occurred during school hours and in a group setting with staff members present between May and June 2024. The questionnaire took approximately 20 min to complete. To ensure clarity and consistency, staff members explained the intent and wording of the questions to participants. The questionnaire avoided technical terms and unfamiliar kanji characters. The school principal, classroom teacher, and nutrition teacher reviewed the questionnaire. Moreover, to avoid missing data, incorrectly completed questionnaires, and entry mistakes, all questions were set as mandatory, and all answers were selected from predefined options. Before full administration, all measures were tested with a small sample of preadolescents and were assessed to be understandable for this age group.

A total of 1525 schoolchildren participated in the survey, but 264 were excluded because of nonresponses to the questions. Hence, the 1261 children who provided valid responses (valid response rate: 82.7%) were included in the analysis. Each dataset was assigned to a unique ID corresponding to demographic characteristics and questionnaire responses. Identifying details were then discarded after the dataset was created and managed using the ID. Participants’ height and weight information were compared to the national average ([36]) using Student’s *t*-test. No significant differences in height and weight were found for either sex.

This study received ethical approval from the Ethics Committee of the Faculty at the university.

### 2.2. General Situation of the Study Area

This study area covered the Kenkyu-gakuen district of Tsukuba City, Ibaraki Prefecture, Japan, in southwestern Ibaraki, approximately 50 km southwest of the prefectural capital and 50 km northeast of Tokyo ([67]). Its population is 241,656 ([21]), and it has the highest population growth rate in Japan (2.3%) ([37]). The second school was in Fukuoka City, the capital and largest city of Fukuoka Prefecture, located in the western Kyushu region ([13]). Fukuoka City has a population of 1,658,999 and 890,695 households ([14]). Among ordinance-designated cities, it ranks first in the population growth rate and the proportion of young people in their teens and twenties ([13]).

Both schools have well-developed information and communication technology (ICT)-based educational systems that incorporate tablet-based learning, digital textbooks, and online exchanges.

### 2.3. Measures

#### 2.3.1. Demographic Characteristics

Participants’ age, sex, standing height, and weight were recorded. Age, sex, height, and weight measurements were obtained from health checkups conducted in April 2024, and the obesity index ([22]) was calculated using these values. Additionally, physical fitness scores were collected from the results of the school’s new physical fitness test, administered during the same period. Overweight and underweight were determined using the measured height and weight.

In Japan, the percentage of overweight (POW), a metric more commonly used for children than BMI-for-age percentile, is calculated based on the measured weight and standard weight-for-height as follows: POW (%) = 100 × (measured weight − standard weight-for-height)/standard weight-for-height. The standard weight-for-height was defined based on age- and sex-specific weight-for-height data from the 2024 Annual Report of School Health Statistics from the [36] ([36]). Japanese standard weight was calculated using an equation derived from height and weight distributions at each age ([42]). The normal POW range for school-age children was between −20% and 20%. POWs of between −<30% and −<20%, and −≤ 30% were defined as moderate and severe thinness, respectively. POWs of between ≥20% and <30%, between ≥30% and <50%, and ≥50% were defined as mild, moderate, and severe obesity, respectively.

#### 2.3.2. SNS Use

The purpose of SNS use was assessed by asking participants: “Please select your most frequently used media service at home (excluding study time, such as homework).” The response options included calling, messaging/chatting (e.g., LINE, KakaoTalk), watching TV, playing games, watching videos (e.g., YouTube), using apps (e.g., Instagram, X, Snapchat, Facebook), searching for information, watching cartoons, reading fiction, and others. Participants were then divided into two groups: SNS users (LINE, KakaoTalk, Instagram, X, Snapchat, and Facebook) and non-SNS users.

#### 2.3.3. Screentime

Daily recreational screen-based activities, defined as media use for non-study purposes, was measured based on the Japanese Ministry’s annual lifestyle habits survey for elementary school children ([35]). Participants were asked the question, “How many hours per day do you usually spend using TV, smartphones, tablets, game consoles, etc. at home? (excluding study time such as homework).” Answers were collected separately for weekdays and weekends using the following categories: “less than 1 h,” “1–2 h,” “2–3 h,” and “more than 3 h.” Average daily screentime was calculated using the following: (average weekday screen time × 5) + (average weekend screen time × 2)/7.

#### 2.3.4. Body Size Perception and Preference

Body image was assessed by silhouette charts ([5]) to maintain children’s engagement and facilitate responses. The charts contained seven silhouettes, ranging from very thin (a value of 1) to very obese (a value of 7). The Collins figures with facial features were performed by presenting the figures in order from 1 to 7. Participants were asked to select the silhouette that best represented their current body image (CBI) and IBI. The difference between CBI and IBI was calculated as the body size perception and preference (CBI − IBI). A body size perception and preference score ≥ 1 indicated that participants’ “desired to be thinner”; a score < 1 indicated participants’ “desired to be heavier”; and a score of 0 indicated that participants were satisfied with their bodies. Body size perception and preference has been validated for elementary school boys and girls ([66]; [7]).

#### 2.3.5. Body Size Misperception

To assess body size misperception, participants categorized their body shape on a five-point scale: thin, slightly thin, normal, slightly overweight, or overweight. Body size misperception was calculated by comparing perceived body shape with actual body shape ([72]). A body size misperception score < 1 was indicated underestimation, meaning a child with a normal body shape perceived themselves as thin/slightly thin or an overweight child perceived themselves as thin/slightly thin/or normal. A distortion score ≥ 1 was indicated overestimation, meaning an underweight child perceived themselves as normal/slightly overweight/ overweight, and normal body shape perceived as slightly overweight/overweight. A distortion score of 0 indicated no distortion, meaning that perceived body shape matched actual body shape. Body size misperception was previously validated in young adult populations ([72]). In the present study, internal consistency was adequate (Cronbach’s α = 0.93 girls, 0.91 boys).

#### 2.3.6. Ideal Body Image

Participants identified their IBI reference group based on relational closeness and included nobody, family members, close friends (i.e., those they regularly interact with), classmates (i.e., those they do not regularly interact with), and media figures (e.g., celebrities, models, idols, athletes, influencers, others on SNS). These classifications are those followed by [60] ([60]). Participants were asked to identify their ideal body image reference from the following options: “Nobody,” “Family,” “Close friends,” “Classmates,” or “Media figures.” To transform this into an analyzable variable, we first set “Nobody” as the reference category (coded as 0). Each of the other responses (“Family,” “Close friends,” “Classmates,” and “Media figures.”) was recoded into a separate binary variable, with 1 indicating selection of that category and 0 otherwise. This approach allowed us to include these variables as a dependent variable in the logistic regression analyses. Internal consistency in this study was high (Cronbach’s α = 0.97 girls, 0.96 boys).

### 2.4. Data Analysis

Sex was dummy-coded (boy = 0, girl = 1), and missing data were removed using the listwise option. Group comparisons were conducted using Student’s *t*-test, the Mann–Whitney U, or Chi-squared tests, as appropriate. Shapiro–Wilk tests confirmed that all variables except weight, height, and POW were non-normally distributed (all *p*’s < 0.001). Therefore, Mann–Whitney U tests were used to compare sex differences and SNS users vs. non-SNS users. As most items exhibited group sex-based differences, all analyses accounted for sex. Hierarchical multivariate linear regression analyses were used to examine whether SNS use was uniquely associated with body size perception and preference and body size misperception. Multinomial logistic regression was conducted to assess the relationship between SNS use and IBI. The dependent variables included body size perception and preference, body size misperception, and IBI (criterion: nobody). The independent variable was SNS use (criterion: do not use). Because screen time, grade, and POW were considered as potentially confounding variables in the multivariate analysis ([29]), Model 1 used screen time, Model 2 used grade, and POW was added to Model 1 as an adjusting variable. The standardized coefficient (β) was calculated for hierarchical multivariate linear regression analyses. Conversely, the odds ratio was determined for multinomial logistic regression. Both analyses used 95% confidence intervals (95% CIs).

In hierarchical multivariate linear regression, assumption testing indicated non-normally distributed residuals in each model. Consequently, results were interpreted using nonparametric bootstrapping. No multicollinearity issues were detected (as all tolerance values were >0.87). For multinomial logistic regression, the Hosmer–Lemeshow was utilized to assess the model’s goodness-of-fit. A priori power calculations were not conducted. However, a post hoc power analysis indicated sufficient power to detect significant associations when three covariates were included (hierarchical multivariate linear regression analyses: boy and girl, 0.99; multinomial logistic regression analyses: boy and girl, 0.99). The power of the unadjusted model was 0.83 for boys and 0.98 for girls.

All statistical analyses were conducted using the Statistical Package for the Social Sciences (SPSS) version 29 (IBM Inc., Tokyo, Japan). The significance level was set to α = 0.05.

## 3. Results

### 3.1. Participant’s Characteristics

The study included 611 boys and 650 girls. Compared to boys, girls more frequently used SNSs, especially LINE and KakaoTalk. Girls also reported a greater body size perception and preference and a stronger IBI. Conversely, boys had longer screen time and a greater body size misperception than girls (Table 1).

### 3.2. Characteristics of SNS Users and Nonusers

SNS users were compared to nonusers regarding grade, obesity index, screen time, and body image constructs by sex. There were no group differences in both sexes’ CBI, IBI, and body size perception and preference. Conversely, compared to the nonuser group, SNS users increased in grade level and had longer screen time. Body size misperception was less in SNS users than nonusers in girls. SNS users recorded less nobody values in IBI and more media figures than the nonuser group. This significant difference was not present in boys (Table 2).

### 3.3. Associations of Body Image with SNS Use

After adjustments for grade, obesity index, and screen time, it was found that SNS use was not significantly associated with body size perception and preference or body size misperception in boys. Because of the IBI, SNS use was associated with close friends and media of IBI when adjusting for screen time. Furthermore, SNS use was not associated with close friends after adjusting for grade, obesity index, and screen time. Conversely, media figures remained a significant predictor of SNS use.

After adjustments for grade, obesity index, and screen time, SNS use was found not to be associated with body size perception and preference in girls. However, when including screen time, SNS use did not significantly correlate with body size misperception, but this relationship disappeared after adjusting for grade and POW. The IBI result showed that SNS use was associated with close friends and media figures when including screen time. Furthermore, similar to boys, SNS use by girls was not associated with close friends after adjusting for grade and POW, while media figures remained a statistically significant predictor of SNS use (Table 3).

## 4. Discussion

While studies continue to accumulate suggesting that SNS use poses a risk to children’s body image, the majority of these studies have been conducted with late adolescents and young adults. This study examined the relationship between SNS use and body image in preadolescents.

The findings indicate that SNS use does not directly contribute to body size perception and preference in Japanese preadolescents. These findings contradict prior results from an Australian sample of girls aged 10–12, wherein girls who reported using SNSs reported higher body surveillance, greater body size perception and preference, more dieting behaviors, and a greater thin-ideal internalization compared to nonusers ([65]). Similarly, a study on preadolescent US girls reported that appearance-focused SNS use exacerbated negative body image concerns ([34]). Furthermore, prior studies on adolescents have identified moderate correlations between SNS use and body satisfaction ([24]; [26]). SNSs serve as a primary source of appearance-related pressure, influencing body esteem through thin-ideal internalization and body comparison, both of which worsen body size perception and preference in adolescents ([54]). Moreover, adolescents encounter various developmental challenges during early to middle adolescence ([55]). Adolescence is a critical period for social comparison and body image sensitivity, and many studies reported that physical comparisons are made with friends and upward social interaction. This results in heightened body size perception and preference.

Body image concerns and body size perception and preference are independent of time spent on SNSs, browsing friends’ SNSs profiles, viewing users’ photos and statuses, and liking and commenting among male and female American college students ([28]), and Instagram among young females ([47]), which are related to the purpose of SNSs and the activities performed on them. However, the absence of a significant association in this study suggests that the purpose of SNS use and the content they are exposed to may not shape body ideals as strongly in preadolescents compared to in adolescents and that they may respond differently to appearance activities on social media concerning their body size perception and preference. For example, children may prefer platforms wherein physical comparisons are less likely to be made with the child’s favorite animal, character, food, or game-related content. Regardless, preadolescents need to establish a correct body image for themselves before puberty to reduce body size perception and preference, which is likely to occur with age. Regardless, most studies on SNS use and body image have predominantly focused on girls, adolescents, and young adults. Limited studies have examined the effects of SNS use on boys or the potential effects of different SNS platforms or cultural variations. While this study included boys in childhood, further investigation remains necessary to understand this issue.

We found that SNS use was significantly associated with body size misperception in girls but not in boys. Girls who used SNSs exhibited higher levels of body size misperception than nonusers. A study focusing on body size misperception among adolescents showed that both moderate and high levels of smartphone dependency were linked to body size misperception ([3]). Moreover, SNS use affecting body size misperception has been found in preadolescent girls ([9]). Our findings align with previous research targeting preadolescents or adolescents ([19]; [3]; [30]; [25]), suggesting that frequent exposure to diverse body images on SNSs disrupts conventional standards and intensifies self-evaluation distortions. However, this result was neither found in boys nor did it persist in girls after adjusting for grade and the obesity index. This suggests that the influence of SNSs on body image is complex and varies by sex, age, and body type. Mental health-related factors such as depression and anxiety are also known to be significantly associated with body size misperception in 13–17-year-olds ([10]). This effect may intensify as adolescence approaches. However, studies on SNS use and body size misperception remain limited, and the mechanisms involved have not been clarified. A more detailed examination of SNS usage patterns, including frequency and content, is needed to clarify these effects.

This study found who the IBI belongs to in preadolescence, which revealed that SNS use strongly correlated with admiration for media figures in both sexes. This suggests that media influence outweighs other social influences (e.g., family, friends, classmates). Although traditional media studies have known that media including television and magazines shape the IBI both preadolescents and adolescents ([48]; [27]; [32]; [56]), the spread of SNSs may amplify this effect. Unlike those of celebrities and models in traditional media, images on SNSs may appear to be realistic comparisons for adolescent girls ([4]). Furthermore, interestingly, appearance comparisons with others of the same-gender peers rather than with models reinforce the internalization of ideal image standards in female adolescents ([41]). Basically, body comparisons with others and with friends may reinforce the desire to lose weight and increase dieting behavior. These findings highlight that even in childhood media portrayals exert a broader influence on body image than direct peer interactions. Notably, these media-driven ideals perpetuate unrealistic standards and could be one of the factors driving SNS use as children attempt to conform their appearance to these ideals.

This consistent association across sexes underscores the universal impact of media, although the IBI may differ between boys and girls. Both preadolescent and adolescent boys typically aspire to a muscular body shape ([49]; [68]; [53]), whereas preadolescent and adolescent girl idealize a thin, beautiful body shape ([17]; [45]), or an appearance that is both thin and muscular in adolescence ([54]). These sex differences emerge even in childhood, and future research on SNSs and body image should include sex-specific measures of body size misperception, including differences in SNS use content, satisfaction with musculature, and IBI.

While this study contributes to the limited body of research on SNS use and body image in preadolescents, several limitations warrant mention. First, the cross-sectional design precludes causal interpretations as establishing temporal relationships is difficult. Future longitudinal studies are necessary to determine whether SNS use predicts body image concerns among children. The findings may also not generalize to other countries and cultures, which have different body ideals and risks to preadolescents’ health behaviors. It will be interesting to examine the effect of cultural differences on preadolescents’ body image in further research.

As a limitation in measurement, the classification of SNS use was binary, and differences based on the detailed content types of SNSs could not be examined. The present study indicates that 36.5% of children used SNSs; however, their SNS use likely varied. Some preadolescents used SNSs for visual content (e.g., Instagram and TikTok), for viewing other people’s photos, and for communication, such as for messaging their friends. Thus, the type of SNSs content that directly associates with children’s body image is not clear. Additionally, photo-based SNSs content likely includes both appearance-focused (e.g., selfies) and non-appearance-focused (e.g., pets) posts. Some preadolescents used SNSs for appearance purposes, such as visually oriented social media like Instagram, TikTok, and Snapchat, for viewing other people’s photos, and for communication, such as for messaging their friends. Thus, the type of SNSs content that directly associates with children’s body image is not clear. Future research should examine and differentiate between the photo-based and non-photo-based SNSs content that children share to better understand this issue among preadolescents. Moreover, children’s engagement level was not assessed in the present study. Investigating children’s psychological and social dimensions could provide valuable insights for enhancing educational settings and refining instructional approaches. Therefore, these variables should be included as measurement items in future research.

Despite these limitations, this study provides important insights that will help guide specific groups of children in using SNSs appropriately and reducing body size perception and preference, body size misperception, and excessive weight loss.

## 5. Conclusions

This study could not confirm an association between SNS use and body size perception and preference among children. However, whilst our findings suggest that SNS use was not linked to body size perception and preference, it may contribute to body size misperception in girls. Additionally, for both boys and girls, SNS use was strongly associated with greater body image admiration for media figures than for their immediate friends and classmates. The present study contributes to the understanding of SNS use and body size perception and preference, body size misperception, and IBI in preadolescent boys and girls, offering insights for future work on this crucial aspect of preadolescent body image.

SNS use can negatively impact body size perception and preference in preadolescents, but there is a lack of research enlightening the particular mechanisms that underlie this relationship. Efforts to reduce SNS use even before adolescence will be critical in supporting the development of a healthy body image in future adolescents. Consistently, gaining insight into how SNS use contributes to body image is key for advancing research in this area and informing prevention efforts aimed at reducing body size perception and preference in preadolescents. In Japan, research on the effects of SNSs on body image for children remains in its early stages. We hope that future research will determine the extent to which engaging with body image content on SNSs can affect body image in children.

## Figures and Tables

**Table 1 ejihpe-15-00125-t001:** Participants characteristics.

Physical Characteristics	Boy (*n* = 611)	Girl (*n* = 650)	*p*
Grade 3	*n* = 137	*n* = 164	
Height (cm) ^1^	128.45 (5.28)	127.58 (5.97)	0.080
Weight (kg) ^1^	26.96 (4.48)	26.72 (4.20)	0.958
POW ^1,4^	−0.89 (11.97)	0.77 (11.17)	0.098
Grade 4	*n* = 179	*n* = 161	
Height (cm) ^1^	132.7 (6.73)	133.20 (7.18)	0.959
Weight (kg) ^1^	30.19 (5.82)	29.09 (5.02)	0.059
POW ^1,4^	0.77 (16.80)	−1.64 (14.81)	0.156
Grade 5	*n* = 162	*n* = 155	
Height (cm) ^1^	140.17 (6.83)	140.33 (7.92)	0.919
Weight (kg) ^1^	35.80 (7.51)	33.94 (7.25)	**0.018**
POW ^1,4^	2.45 (14.90)	−0.63 (14.14)	0.055
Grade 6	*n* = 133	*n* = 170	
Height (cm) ^1^	146.37 (8.36)	147.95 (6.03)	**0.009**
Weight (kg) ^1^	40.44 (10.38)	39.79 (6.37)	0.391
POW ^1,4^	2.94 (20.01)	0.24 (13.91)	0.392
SNSs	Boy (*n* = 611)	Girl (*n* = 650)	*p*
Used SNSs ^3^			**<0.001**
Yes	181 (29.6)	279 (42.9)	
No	430 (70.4)	371 (57.1)	
Contents ^3^			
LINE, Kakao Talk			**<0.001**
Yes	165 (27.0)	265 (40.8)	
No	446 (73.0)	385 (59.2)	
Instagram, Snapchat			0.233
Yes	3.9 (24)	5.4 (35)	
No	96.1 (587)	94.6 (615)	
X, Facebook			1.000
Yes	3.1 (19)	3.2 (21)	
No	96.9 (592)	96.8 (629)	
Screentime (minutes) ^2^	98.31 (64.29,132.86)	88.02 (47.14,115.71)	**<0.001**
Body Image	Boy (*n* = 611)	Girl (*n* = 650)	*p*
CBI ^2,5^	3.89 (3.00,4.00)	3.83 (3.00,4.00)	0.100
IBI ^2,5^	3.77 (3.00,4.00)	3.45 (3.00,4.00)	**<0.001**
Body size perception and preference ^2,6^	0.12 (0.00,1.00)	0.38 (0.00,1.00)	**<0.001**
Body size misperception ^2,7^	−0.31 (−1.00,0.00)	−0.18 (−1.00,0.00)	**0.007**
Ideal body image ^3^			**<0.001**
Nobody	426 (69.7)	393 (60.5)	
Family	19 (3.1)	28 (4.3)	
Close friends	38 (6.2)	69 (10.6)	
Classmates	8 (1.3)	28 (4.3)	
Media figures	120 (19.6)	132 (20.3)	

Note. Abbreviations: POW, percentage of overweight; SNSs, social networking services; CBI, current body image; IBI, ideal body image. ^1^ Values represent mean (SD). *p*-values were analyzed via Student’s *t*-test. ^2^ Values represent mean (IQR). *p*-values were analyzed via the Mann–Whitney U test. ^3^ Values represent n (%). *p*-values were analyzed by the Chi-square test. ^4^ POW (calculated from the standard weight by sex, age, and height used in the School Health Statistics Survey). ^5^ The score range is 1–7 (1 = very thin, 7 = very obese). ^6^ Body size perception and preference represents the difference between CBI and IBI (CBI−IBI). ^7^ The score range is −4–4 (<1 = underestimation, 0 = no distortion, >1 = overestimation). Bold: significant differences in *p*-Value.

**Table 2 ejihpe-15-00125-t002:** Characteristics of social networking service users and nonusers.

Item	Boy (*n* = 611)			Girl (*n* = 650)		
	SNS Users	Non-SNS Users	*p*	SNS Users	Non-SNS Users	*p*
Grade ^3^						
3	16(8.8)	125(29.1)	**<0.001**	43(15.4)	122(32.9)	**<0.001**
4	45(24.9)	131(30.5)		55(19.7)	102(27.5)	
5	56(30.9)	102(23.7)		78(28.0)	81(21.8)	
6	64(35.4)	72(16.7)		103(36.9)	66(17.8)	
POW ^1,4^	2.70(14.78)	0.55(17.24)	0.146	−0.08(12.80)	−0.71(13.21)	0.543
Screentime (minutes) ^2^	106.95(72.86,153.21)	94.73(64.29,115.71)	**0.002**	106.95(47.14,115.71)	94.73(47.14,115.71)	**0.025**
CBI ^2,5^	3.99(3.00,5.00)	3.85(3.00,4.00)	0.090	3.83(3.00,4.00)	3.82(3.00,4.00)	0.805
IBI ^2,5^	3.78(3.00,4.00)	3.77(3.00,4.00)	0.900	3.78(3.00,4.00)	3.77(3.00,4.00)	0.961
Body size perception and preference ^2,6^	0.22(0.00,1.00)	0.08(−1.00,1.00)	0.159	0.22(0.00,1.00)	0.08(0.00,1.00)	0.927
Body size misperception ^2,7^	−0.33(−1.00,0.00)	−0.36(−1.00,0.00)	0.138	−0.20(0.00,0.00)	−0.36(−1.00,0.00)	**0.014**
Ideal body image ^3^						
Nobody	114(63)	312(72.6)	0.109	150(53.8)	243(65.5)	**0.004**
Family	5(2.8)	14(3.3)		14(5)	14(3.8)	
Close friends	16(8.8)	22(5.1)		36(12.9)	33(8.9)	
Classmates	2(1.1)	6(1.4)		8(2.9)	20(5.4)	
Media figures	44(24.3)	76(17.7)		71(25.4)	61(16.4)	

Note. Abbreviations: POW, percentage of overweight; SNSs, social networking services; CBI, current body image; IBI, ideal body image. ^1^ Values represent mean (SD). *p*-values were analyzed via Student’s *t*-test.^2^ Values represent mean (IQR). *p*-values were analyzed via the Mann–Whitney U test. ^3^ Values represent n (%). *p*-values were analyzed by the Chi-square test. ^4^ POW (calculated from the standard weight by sex, age, and height used in the School Health Statistics Survey). ^5^ The score range is 1–7 (1 =very thin, 7 =very obese). ^6^ Body size perception and preference represents the difference between CBI and IBI (CBI−IBI). ^7^ The score range is −4–4 (<1 = underestimation, 0 = no distortion, >1= overestimation). Bold: significant differences in *p*-value.

**Table 3 ejihpe-15-00125-t003:** Associations of body image with social networking services use.

Outcomes Dependent Variables	Univariate Analysis		Multivariate Analysis (Model 1)		Multivariate Analysis (Model 2)	
	β/OR [95%CI]	SE	β/OR [95%CI]	SE	β/OR [95%CI]	SE
Boys (*n* = 611)						
Body size perception and preference ^1^	0.06 [−0.04, 0.29]	0.09	0.05 [−0.06, 0.29]	0.09	0.03 [−0.09, 0.22]	0.08
Body size misperception ^1^	0.07 [−0.01, 0.30]	0.08	0.07 [−0.01, 0.30]	0.08	0.04 [−0.09, 0.24]	0.08
Ideal body image ^2^						
Nobody	1		1		1	
Family	0.98 [0.34, 2.78]	0.53	0.98 [0.34, 2.81]	0.54	1.08 [0.36, 3.20]	0.55
Close friends	2.00 [1.01, 3.92] *	0.35	1.92 [0.97, 3.81]	0.35	1.74 [0.86, 3.55]	0.36
Classmates	0.91 [0.18, 4.59]	0.82	1.10 [0.22, 5.58]	0.83	1.52 [0.27, 8.44]	0.87
Media figures	1.58 [1.03, 2.43] *	0.22	1.71 [1.11, 2.65] *	0.22	1.64 [1.04, 2.57] *	0.23
Girls (*n* = 650)						
Body size perception and preference ^1^	0.01 [−0.13, 0.16]	0.07	0.00 [−0.14, 0.15]	0.07	−0.01 [−0.15, 0.13]	0.07
Body size misperception ^1^	0.09 [0.02, 0.28] *	0.07	0.08 [0.00, 0.26] *	0.07	0.04 [−0.07, 0.20]	0.07
Ideal body image ^2^						
Nobody	1		1		1	
Family	1.62 [0.75, 3.49]	0.39	1.60 [0.74, 3.47]	0.39	1.93 [0.87, 4.30]	0.41
Close friends	1.77 [1.06, 2.96] *	0.26	1.74 [1.04, 2.92] *	0.26	1.56 [0.91, 2.67]	0.27
Classmates	0.65 [0.28, 1.51]	0.43	0.64 [0.28, 1.50]	0.43	0.61 [0.25, 1.45]	0.45
Media figures	1.89 [1.27, 2.81] **	0.20	1.87 [1.25, 2.78] *	0.20	1.72 [1.14, 2.60] *	0.21

Note. N = 1261. * *p* < 0.05, ** *p* < 0.01. Abbreviations: SNSs, social networking services; β, standardized coefficient; OR, odds ratio; CIs, confidence intervals; SE, standard error. β/OR, the results of the β calculation using continuous variables (body size perception and preference and body size misperception) and the OR calculation using binary dependent variables (ideal body image). ^1^ Values represent β [95% CI]. *p*-values were analyzed via linear regression analysis. β values are standardized beta values from the non-bootstrapped model; SE and CI are from the bootstrapped model. ^2^ Values represent OR [95% CI]. *p*-values were analyzed via multinomial logistic regression analysis. Model 1: Effect estimates adjusted for screen time. Model 2: Model 1 + grade and POW. Ideal body image criterion: Nobody.

## Data Availability

Restrictions apply to the availability of these data due to the inclusion of sensitive or confidential information such as the height and weight of prepubescent children. Data were obtained from Gakuen no Mori Compulsory Education School and are available from the authors with the permission of Gakuen no Mori Compulsory Education School.

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
