# Peer review of "Association Between Social Networking Service Use and Body Image Among Elementary School Children in Japan"

_ejihpe, 2025, doi:10.3390/ejihpe15070125_

Round 1

Reviewer 1 Report

Comments and Suggestions for Authors

Dear Authors,

Thank you for your manuscript. Please see my comments below.

This is a well-written and timely study addressing an important public health issue: the relationship between social networking service (SNS) use and body image in preadolescent children. The sample is large and includes both boys and girls, with careful attention to ethical data collection and age-appropriate instruments. However, several revisions are needed to improve clarity, rigor, and consistency across sections.

  1. Abbreviations like "SNS" should be avoided in the title. Please spell out.

  2. The abstract should state the study aim, as it currently jumps directly into findings. Also, include participant demographics (sex distribution and age range) for better context. Provide brief details on the methods used to measure key variables.

  3. The study aim should be stated in section 1.6 (Current study), not just the hypotheses. Also, consider summarising how this study addresses gaps in prior literature more succinctly.

  4. The Silhouette Scale is inaccurately described as measuring body dissatisfaction (section 2.3.4). It assesses body size perception and preference. Please revise the term accordingly. Examples of body dissatisfaction assessment can be checked from the Eating Disorder Inventory 2 (EDI-2) eating-disorder-specific scales or the opposite construct, Body Area Satisfaction, from the MBSRQ-AS.

  5. The term “body cognitive distortion” (section 2.3.5) is uncommon and inconsistently used (further, the authors use the term "body perception"). A more accurate description would be “inaccuracy in body weight estimation” or “body size misperception,” aligning with existing literature. Ensure consistent terminology throughout the paper.

  6. The construction of the Ideal Body Image (IBI) variable is not well explained (section 2.3.6). Clarify how responses were transformed into an analyzable variable, especially since it becomes a dependent variable in logistic regression.

  7. Avoid using abbreviations (e.g., “SNS”) in table titles (Table 2). 
  8. The heading “Table 3. Associations of body image with SNS use This is a table.” is unprofessional and should be corrected. The dependent variables in Table 3 are not labelled in the title or clearly described in the statistical analysis section. The use of “β/OR” in the table is confusing and requires an additional explanation because OR calculation requires a dichotomous dependent variable, while β normally distributed continuous variable.

  9. In the Discussion, when citing earlier studies, clarify the methodological differences (e.g., adolescent vs. preadolescent samples) more explicitly.
  10. The study correctly notes its cross-sectional design, but it would benefit from also noting the limitations in measurement, especially the binary SNS use classification and the absence of detailed content type or engagement level.

Author Response

Please see the attachment." in the box if you only upload an attachment.

Reviewer 2 Report

Comments and Suggestions for Authors

This is a well-written and interesting manuscript that utilizes a large sample size of pre-adolescent participants to assess the relationships between the use of social networking services and body image. In general, the article can be of interest to a wide audience, especially researchers in the field of mental health and auxology. The content is original, and the discussion is well-argued. However,  I found some weaknesses.

My suggestions for improving the manuscript primarily concern methodological aspects:

  • In the Introduction (section 1.4), it is not clear what the boys' main concern is. Although this topic is taken up in the discussion, it is necessary to supplement this part by referring to their prevailing ideal of body muscularity.
  • In the Materials and Methods (section 2.3.1), it is necessary to clarify whether the data used in the study refer to the weight and stature values self-reported by the children or those measured directly through health checkups.
  • At line 211, you cite an obesity index. It needs to be clarified what you mean and whether what is reported at lines 216 et seq. is referable to it.
  • In the Materials and Methods (section 2.3.4), it is not clear whether the administration of the Collins figures was done by presenting the figures in a casual order or ordered from 1 to 7. Also, many researchers eliminate facial features of the silhouettes before submitting the pictures so as not to affect the choice in any way. What was done?
  • In the Materials and Methods (section 2.4), it is surprisingly stated that “all variables were non-normally distributed,” whereas the normal distribution of weight and stature is an acquired fact in Anthropometry.
  • The Conclusions are to be revised by removing citations from the literature, which traditionally should be included in the introduction and/or discussion.

Minor concerns: 

  • Line 210: Change “Age, sex, height, and weight” to “Age, sex, standing height, and weight”.
  • Table 1: Add the unit of measurement to the variable Screentime.
  • Table 2: Adjust the numbers in the Body dissatisfaction row.
  • Table 3: Delete the words “This is a table”.
  • Line 419: Replace ‘they’ with “the girls.”

Author Response

(The authors gave the same response as above.)

Round 2

Reviewer 1 Report

Comments and Suggestions for Authors

Dear Authors,
I believe the revised version of your paper is significantly improved and much clearer. Well done!